# The Diagnostic Efficacy of and Requirement for Postnatal Ultrasonography Screening for Congenital Anomalies of the Kidney and Urinary Tract

**DOI:** 10.3390/diagnostics13193106

**Published:** 2023-09-30

**Authors:** Abdulgani Gulyuz, Mehmet Tekin

**Affiliations:** 1Department of Pediatrics, School of Medicine, Turgut Ozal University, 06560 Malatya, Turkey; abdulganigulyuz@gmail.com; 2Department of Pediatrics, School of Medicine, Inonu University, 44280 Malatya, Turkey

**Keywords:** children, congenital anomalies of the kidney and urinary tract, prenatal, postnatal, ultrasound screening

## Abstract

Background: We aimed to investigate the efficacy of postnatal ultrasonography in detecting congenital anomalies of the kidneys and urinary tract in term infants without prenatal history of congenital anomalies of the kidneys and urinary tract. Methods: In this retrospective cohort study, we reviewed the records of term infants between six weeks and three months of age who underwent urinary tract ultrasonography during routine pediatric care. Results: Congenital anomalies of the kidneys and urinary tract were detected on prenatal ultrasonography in 75 of the 2620 patients included in the study. Congenital anomalies of the kidneys and urinary tract were detected via postnatal USG in 46 (1.8%) of 2554 patients without anomalies on prenatal USG screening. The most common anomaly was hydronephrosis (69.6%). Thirty-two cases of hydronephrosis, three cases of renal agenesis, four cases of horseshoe kidney, one case of MCDK, and two cases of duplex systems which were not detected on prenatal USG were detected on postnatal USG. On the other hand, 29 (1.1%) cases with mild or moderate hydronephrosis on prenatal ultrasonography did not have hydronephrosis on postnatal ultrasonography. Conclusions: In our study, approximately one-third of the cases of hydronephrosis, unilateral renal agenesis, duplex systems, horseshoe kidney, and ectopic kidney were not detected in prenatal ultrasonography screening. Therefore, we believe that in addition to prenatal ultrasonography screening, postnatal ultrasonography screening of all children for urinary tract anomalies would be beneficial.

## 1. Introduction

Congenital anomalies of the kidney and urinary tract (CAKUT) are still the most frequent cause of chronic kidney failure [1]. Since CAKUT can be asymptomatic and can lead to hypertension and end-stage renal failure in later years, many countries try to detect cases prenatally [2,3,4,5]. Ultrasonography (USG) is preferred as a screening method because it is inexpensive, non-invasive, reproducible, and reliable [6,7].

The reported incidence of CAKUT in prenatal USG screening is 0.3–5% [2,3,4,5]. However, resolution is observed in a significant proportion of these cases on imaging performed within the first postnatal week [8,9]. On the other hand, it has been reported that approximately 10–20% of hydronephrosis cases are determined in the postnatal period, and some collecting system anomalies, hydroureter, and pelvic kidney cannot be detected on prenatal USG [9]. These inconsistent findings have been attributed to physiological variations between the fetus and infant and positional variation during USG imaging. Therefore, postnatal USG is reported to be more effective in detecting CAKUT than prenatal USG [8,9,10].

Postnatal USG screening is generally performed in order to confirm diagnosis when CAKUT is detected prenatally. Since CAKUT is a preventable cause of end-stage renal disease, public health procedures are needed to identify cases of CAKUT that are not detected prenatally due to technical limitations. To the best of our knowledge, there are few postnatal USG screening studies in infants without a history of prenatal CAKUT [11,12,13,14,15]. The aim of this study was to investigate the effectiveness of postnatal USG in detecting CAKUT in term infants without a history of CAKUT in the prenatal period.

## 2. Methods

### 2.1. Study Population

In this retrospective cohort study, the records of infants aged six weeks to three months who were admitted to Malatya Sevgi Hospital for routine pediatric care and underwent urinary tract USG between 1 January 2010 and 31 December 2017 were reviewed. The frequency and importance of urinary system anomalies were explained to the families, and the cases who accepted ultrasonographic examination were included in the study. The study was conducted in accordance with the principles of the Declaration of Helsinki, approved by the local ethics committee (No. 2018/06), and informed consent was obtained from all parents.

Term infants born between 37 and 42 weeks were included in the study. Patients with prematurity, external genital anomalies such as hypospadias, multiple congenital anomalies, and a familial history of congenital urinary anomalies or urinary tract infections were excluded.

### 2.2. Equipment and Procedures

All USG scanning procedures were performed by the same radiologist experienced in pediatric urology using a TOSHIBA Aplio 400 device (Toshiba Medical Systems Corporation, Otawara, Japan) and a 5 MHz convex and 8 MHz linear probe. Children were well-hydrated and examined in the prone position before scanning. The anteroposterior diameter (APD) of the renal pelvis was measured in the transverse plane. The kidneys were evaluated for shape, size, location, parenchymal thickness, echogenicity, cysts, and calyceal dilatation. Dilated ureters, and bladder diameter and wall thickness were also evaluated.

### 2.3. Definitions

Hydronephrosis, renal agenesis, renal hypoplasia, horseshoe kidney, ectopic kidney, multicystic dysplastic kidney (MCDK), duplex system, posterior urethral valve (PUV), ureteropelvic junction obstruction (UPJO), and vesicoureteral reflux (VUR) were classified as CAKUT. Ureterocele, duplex kidney, PUV, VUR and UPJO were defined as “major urologic pathology” in cases of hydronephrosis.

Hydronephrosis was defined according to APD measurement (7–9 mm, mild; 9–15 mm, moderate; >15 mm, severe) [6]. Resolution was defined as an APD < 7 mm at two consecutive USG imaging sessions, and normal renal parenchyma, calyces, ureters, and bladder.

Voiding cystourethrography (VCUG) and 99mTc-dimercaptosuccinic acid scintigraphy (DMSA) were performed in cases with unilateral or bilateral APD > 15 mm, ureteral dilatation, febrile urinary tract infection during follow-up, pelvic dilatation persisting for more than 12 months, and/or increasing dilatation. Diethylenetriamine pentaacetic acid (DTPA) scintigraphy was performed in all cases with severe hydronephrosis or ureteral dilatation without VUR [7].

VUR was staged according to the International Reflux Study Committee Classification [16]. UPJO was defined as the absence of dilatation of the ureters despite the renal pelvis exceeding 10–15 mm, accompanied by calyceal dilatation, and abnormal diuretic renogram pattern. In DTPA imaging, a diuretic was administered at the twentieth minute when the activity of the collecting system was highest. After diuretic administration, the time until the activity of the collecting system decreases by 50% is called the T1/2 time. A T1/2 time > 20 min was considered an abnormal diuretic renogram pattern [17]. MCDK was defined as the presence of multiple, variably-sized, non-communicating cysts in the absence of normal renal parenchyma, and duplex system was defined as a single renal unit drained by two collecting systems [6].

Urinary tract infection (UTI) was defined as the growth of a single pathogen at a level of >100,000 CFU/mL, with fever (>38 °C), presence of urinary symptoms, and/or leukocyturia in samples collected with sterile urine bags [18].

### 2.4. Follow-Up

CAKUT cases were followed for 6–36 months, during which time USG, urine cultures, blood pressure measurements, and growth measurements were performed every three months. Proteinuria and estimated glomerular filtration rate (eGFR) were evaluated in patients with persistent anomalies; eGFR was calculated from plasma creatinine using the Schwartz formula. Impaired renal function was defined as eGFR < 90 mL/min per 1.73 m^2^ according to the KDIGO guidelines for stages 2–5. End-stage renal failure (ESRF) was defined as being on renal replacement therapy [19]. Prophylactic antibiotic treatment was initiated in patients with severe unilateral or bilateral hydronephrosis (APD > 15 mm) or a dilated ureter, in patients with febrile UTI during follow-up, and in patients with VUR or MCDK until VCUG was performed.

Surgical interventions were performed in patients with persistent grade IV–V VUR after the age of one year, in patients with febrile UTI despite prophylaxis or new scar formation in the renal parenchyma, in patients with scintigraphic function of less than 40% in the ipsilateral kidney, in UPJO patients with more than 10% deterioration in renal function or decrease in parenchymal thickness during follow-up, and in patients with a duplex system with concomitant unilateral obstruction (ureterocele or ectopic ureter).

### 2.5. Data Collection

Demographic data, including sex, gestational age at birth, prenatal USG imaging, history of UTI, family history of CAKUT, and laboratory results, including USG, DMSA, DTPA, VCUG, complete urinalysis and urine culture, final diagnosis, whether UTI developed or not, duration of resolution, and surgical procedures performed, were recorded.

### 2.6. Statistical Analysis

Data were analyzed using SPSS 23.0 software (SPSS, Inc., Chicago, IL, USA). Descriptive statistics were analyzed. The chi-square test was used to analyze categorical variables. *p* values < 0.05 were considered statistically significant.

## 3. Results

The files of 2629 cases, 1218 (46.5%) girls and 1402 (53.5%) boys, who underwent postnatal USG screening were analyzed. The cases were divided into two groups: those with CAKUT detected on prenatal USG scan and those with CAKUT detected only on postnatal USG scan.

### 3.1. CAKUT Detected on Prenatal USG

There were 75 cases with CAKUT detected via prenatal USG screening. Hydronephrosis was detected in fifty-two cases, renal agenesis in six, horseshoe kidney in three, MCDK in three, ectopic kidney in three, and polycystic kidney in one case (Table 1). On the other hand, 29 patients with mild or moderate hydronephrosis on prenatal USG had no hydronephrosis on postnatal USG scans.

### 3.2. CAKUT Detected Only on Postnatal USG Scan

In this group, which included 2554 cases without CAKUT on prenatal USG screening, CAKUT was detected in 46 (1.8%) patients on postnatal USG screening. Of the CAKUT cases, 33 (71.7%) were boys and 13 (28.3%) were girls, with a boy:girl ratio of 2.5:1 (*p* < 0.001). The most common anomaly was hydronephrosis (32 cases, representing 69.6% of CAKUT cases, and 1.25% of all cases). Hydronephrosis was more common in boys, with a male:female ratio of 2.7:1. Mild dilatation was present in twenty-two (68.8%) cases of hydronephrosis, moderate dilatation in seven, and severe dilatation in three. In addition, renal agenesis was detected in three cases, horseshoe kidney in four, echogenic kidney in one, MCDK in one, ectopic kidney in two, simple cyst in one, and duplex system in two cases (Table 1).

### 3.3. Outcomes of Cases Detected on Prenatal USG

Spontaneous resolution was observed in 34 (65.4%) cases of hydronephrosis detected on prenatal USG. While the degree of hydronephrosis did not change on renal USG during follow-up in four cases of moderate hydronephrosis, an increase was detected in one mild and two moderate cases. VUR was diagnosed in two of the patients in the group with no change in the degree of hydronephrosis, while UPJO was detected in three patients in this group with increased hydronephrosis. UTI was diagnosed in 19 (36.5%) patients during follow-up. VCUG imaging was performed in 29 cases, and DTPA imaging in 20 cases. Major urologic pathology was detected in a total of 18 (34.6%) cases, 10 of which were VUR and 8 of which were UPJO. One VUR case was stage I, two were stage II, two were stage III, three were stage IV, and two were stage V. Surgery was performed in ten patients: six with VUR, and four with UPJO (Table 2).

A UTI developed in one patient diagnosed with horseshoe kidney. Grade III VUR was found in a patient with MCDK with unilateral hydronephrosis, and grade II VUR was found in a patient with horseshoe kidney who developed a UTI. Antihypertensive treatment was started in a patient with unilateral renal agenesis who developed hypertension during follow-up. None of the cases with CAKUT on prenatal USG had deterioration in renal function (eGFR < 90 mL/min/1.73 m^2^), ESRF, or proteinuria during the follow-up period of 6 to 36 months.

### 3.4. Outcomes of Cases Detected on Postnatal USG

Spontaneous resolution was observed in 23 (71.9%) cases of hydronephrosis detected only on postnatal USG. While the degree of hydronephrosis did not change on renal USG during follow-up in two cases of moderate hydronephrosis, an increase was found in one mild case. VUR was detected in one case in the increased hydronephrosis group. UTIs were diagnosed in six (18.8%) cases during follow-up. VCUG imaging was performed in 15 cases and DTPA imaging in 10 cases. Major urologic pathology was detected in five (15.6%) cases, four of which were VUR and one was UPJO. One of the VUR cases was stage I, one was stage III, one was stage IV and one was stage V. Three patients with VUR underwent surgery (Table 2).

One patient with horseshoe kidney and two patients with duplex systems developed UTIs. Grade II VUR was found in one horseshoe kidney patient with a UTI, grade II VUR was found in one of the two duplex system patients with UTIs, and grade III VUR was observed in the other. Surgery was performed in one case of a duplex system associated with ureterocele. In none of the cases in which CAKUT could be detected only on postnatal USG were renal function deterioration (eGFR < 90 mL/min/1.73 m^2^), ESRF, or proteinuria observed during the 6 to 36 months follow-up period.

Detection of urologic pathology (*p* < 0.001) and frequency of surgery (*p* < 0.001) was higher in the severe hydronephrosis group than in the mild and moderate groups.

## 4. Discussion

CAKUT is still the most common cause of end-stage renal failure in childhood [1]. Renal hypoplasia has also been reported to increase the risk of end-stage renal failure in adulthood [20]. Therefore, it is important to detect and, if possible, treat cases of CAKUT early in life. For this purpose, the urinary system is screened prenatally, and cases with CAKUT are followed up with postnatal ultrasonography. However, it is known that some cases of CAKUT cannot be detected using prenatal USG due to technical and physiological reasons [9,13,21,22]. In the few studies investigating the prevalence of CACUT with postnatal USG, there was no standardized screening period; screening was performed over a wide time interval from three days to six months and a CACUT prevalence of 1.5% to 7.4% was reported [11,12,13,14,15]. In this retrospective cohort study in which we screened infants between six weeks and three months of age without a history of CAKUT prenatally with postnatal USG, we found the prevalence of CAKUT to be 1.8%. We believe that this wide variation in the prevalence of CAKUT may be due to the physiologic status of the subjects at the time of screening, and especially whether transient cases of hydronephrosis were detected.

In our study, thirty-two cases of hydronephrosis were detected via postnatal USG, as well as three cases of renal agenesis, four cases of horseshoe kidney, one case of MCDK, and two cases of duplex systems that could not be detected using prenatal USG. Drnasin et al. [21] reported the prevalence of hydronephrosis as 7.4% in a postnatal study of 1000 healthy infants screened with prenatal USG and reported as “normal”. Miyakita et al. [11] reported that 39% of 92 cases of hydronephrosis were detected in the prenatal period. Hálek et al. [13] reported that only 8.5% of 234 cases of hydronephrosis were detected in the prenatal period, and all cases of unilateral agenesis and ectopic kidney were diagnosed on postnatal USG screening. Richter-Rodier et al. [15] reported that only 16.4% of hydronephrosis cases were detected in prenatal screening and that although prenatal USG showed high efficiency in detecting cases of renal agenesis and MCDK, its efficiency in detecting duplex system, horseshoe kidney, dysplastic, and ectopic kidney was low. Mamì et al. [22,23] reported that prenatal USG detected only 35.7% of moderate hydronephrosis cases and 73.2% of severe hydronephrosis cases. In our study, hydronephrosis was detected on prenatal USG screening in 53.2% of mild hydronephrosis cases, 68.2% of moderate cases, and 80% of severe cases. Consistent with previous studies, we observed a high rate of detection of severe hydronephrosis in the prenatal period in our study.

There are also authors who argue that postnatal USG scans have no benefit. Cauilo et al. [24] reported the incidence of urinary pathology detected via postnatal USG as 0.96% and the incidence of cases requiring surgery as 0.24% in a study excluding those diagnosed with CAKUT in prenatal screening, and concluded that postnatal USG screening is unnecessary because most patients requiring surgery would probably be symptomatic and would have sought medical attention without routine screening. Scott et al. [25] also suggested that routine renal ultrasound screening of newborn infants has no value in identifying those who may have ureteric reflux and that the number of other renal abnormalities likely to be detected is small. In our study, the prevalence of CAKUT on postnatal USG screening was 1.8% and the proportion of cases requiring surgery was 0.12%. Screening for urinary anomalies is important not only to detect pathologies requiring surgery but also to prevent hypertension and end-stage renal failure. The gold standard diagnostic method for detecting ureteric reflux is VCUG, but due to high radiation exposure, follow-up is currently performed with USG, and VCUG is only used in selected cases.

Consistent with previous studies, CAKUT was more common in boys and on the left side [11,12,13,14,15]. Hydronephrosis was the most common anomaly (69.6%) and 68.8% of these cases were mild. Hydronephrosis was more common in boys and in the left kidney. Similarly, Hálek et al. [13] and Richter-Rodier et al. [15] also reported that hydronephrosis was the most common anomaly, most of these cases were mild, and hydronephrosis was more common in boys and in the left kidney.

Spontaneous resolution occurred in 65.4% of cases of hydronephrosis detected on prenatal USG, compared to 71.9% of cases detected only on postnatal USG. In studies examining cases detected in postnatal screening, Drnasin et al. [21] reported a total spontaneous resolution rate of 79.2%, including 83.1% of mild cases and 33.3% of moderate-to-severe cases, while Hálek et al. [13] reported 82.1%. In studies examining cases detected in the prenatal period, Barbosa et al. [26] reported spontaneous resolution in 90% and 75% of mild and moderate hydronephrosis, respectively, while Coelho et al. [27] reported spontaneous resolution in 97% and 78% of mild and moderate hydronephrosis, respectively. Consistent with previous postnatal screening studies, we believe that the slightly lower rates of spontaneous resolution compared to prenatally detected cases may be attributed to the fact that more physiologic hydronephrosis is detected in prenatal screening in accordance with fetal physiology. Indeed, deGrauw et al. [5] reported that 24% of antenatal hydronephrosis cases were normal on the first postnatal USG, while Gökçe et al. [28] reported that 9.4% were normal.

VUR was detected in 16.7%, UPJO in 10.7% and surgery was performed in 15.5% of hydronephrosis cases. These rates were 12.5%, 3.1% and 9.4%, respectively, in cases where hydronephrosis could only be detected using postnatal USG. In addition, surgery was performed in one case of duplex system associated with ureterocele. We observed that the frequency of urologic pathology detection and surgical intervention increased with the degree of hydronephrosis. Nef et al. [29] reported that approximately one-third of patients underwent an operation in cases detected in the prenatal period, while Miyakita et al. [11] reported an operation rate of 3.3% in cases detected only in postnatal screening. Bhide et al. [30] reported that 59.6% of patients underwent an operation for CAKUT that could be detected in the prenatal period. Hálek et al. [13] reported that the incidence of surgery was 7.1% in cases with hydronephrosis detected only in postnatal screening and that the rates of urologic pathology detection increased in parallel with APD. Richter-Rodier et al. [15] reported the incidence of surgery as 12.1%. Grazioli et al. [31] reported that prenatal USG did not predict VUR but that APD measured using postnatal USG correlated with the risk of VUR.

None of our patients with CAKUT detected in the prenatal or postnatal period had end-stage renal failure or proteinuria during a follow-up period of six to thirty-six months. In studies examining cases detected in the prenatal period, Nef et al. [29] found ESRF in only 2 of 115 patients during an 18-year follow-up period, while Herthelius et al. [32] reported that none of their patients had ESRF during a 12–15-year follow-up period. Miyakita et al. [11] reported that they did not detect ESRF in any of the patients with CAKUT detected using postnatal USG scanning, independent of prenatal history. Although our follow-up period was short, our data suggest that progression to end-stage renal failure can be prevented in patients with CAKUT who are detected and treated early.

We are aware that our study has some limitations. First, the retrospective nature of our data made the standardization of ultrasonography data difficult, especially since prenatal USG scans are subjective because they were performed in many different centers and by different radiologists, whereas postnatal scans were performed consistently by the same radiologist in our center who is very experienced in this field. Secondly, it is not possible to know the exact number of cases with VUR in our study because VCUG was not performed in all patients but in those who met the specified criteria. Third, it is difficult to determine the clinical significance of the anomalies detected, as low-grade VUR cases resolved spontaneously and many cases of horseshoe kidney were not surgically intervened but only followed-up. However, prevention or early treatment of urinary tract infections as a result of early recognition of mild-to-moderate cases of VUR will prevent damage to the kidney; similarly, we believe that early detection of non-invasive anomalies is important to prevent renal failure and hypertension later in life. Finally, most cases of hydronephrosis are physiologic and resolve spontaneously. In this regard, we cannot provide a clear recommendation on the ideal timing of ultrasonography to detect these physiologic anomalies, which may cause unnecessary anxiety in families, as well as to detect serious anomalies requiring surgery.

## 5. Conclusions

In conclusion, CAKUT, one of the causes of preventable end-stage renal disease, is an important public health problem. Postnatal USG screening is an effective method to identify cases of CAKUT. In our study, approximately one-third of cases of hydronephrosis, unilateral renal agenesis, duplex system, horseshoe kidney and ectopic kidney were not detected in prenatal USG screening. Therefore, in addition to prenatal USG screening, we recommend that all children should be screened with postnatal USG for urinary tract anomalies as a public health practice. Our data should be corroborated with larger patient cohorts, especially the timing of optimal USG screening.

## Figures and Tables

**Table 1 diagnostics-13-03106-t001:** Incidence and characteristics of congenital anomalies of the kidney and urinary tract.

	Number (*n*)		Percent (%)		Gender (F/M)	Affected Side		
						Right (*n*)	Left (*n*)	Bilateral (*n*)
	Prenatal ^#^	Postnatal ^§^	Prenatal ^#^	Postnatal ^§^				
Total CAKUT (*n* = 121)	75	46	62.0	38.0	36/85	17 (14)	78 (64.5)	26 (21.5)
Renal agenesis (*n* = 9)	6	3	66.7	33.3	1/8	4	5	-
Horseshoe kidney (*n* = 7)	3	4	42.9	57.1	3/4	-	-	7
Echogenic kidney (*n* = 7)	5	2	71.4	28.6	5/2	2	1	4
Multicystic dysplastic kidney (*n* = 4)	3	1	75.0	25.0	2/2	1	3	-
Ectopic kidney (*n* = 4)	3	1	75.0	25.0	0/4	2	2	-
Simple cyst (*n* = 3)	2	1	66.7	33.3	1/2	2	1	-
Duplex system (*n* = 2)	0	2	0	100	0/2	1	1	-
Polycystic kidney (*n* = 1)	1	0	100	0	1/0	-	-	1
Hydronephrosis (*n* = 84)	52	32	61.9	38.1	23/61	5 (5.9)	65 (77.4)	14 (16.7)
Mild (7–9.9 mm) (*n* = 47)	25	22	53.2	46.8	12/35	2 (4.3)	42 (89.4)	3 (6.4)
Moderate (10–15 mm) (*n* = 22)	15	7	68.2	31.8	7/15	2 (9.1)	14 (63.6)	6 (27.3)
Severe (>15 mm) (*n* = 15)	12	3	80.0	20.0	4/11	1 (6.7)	9 (60.0)	5 (33.3)

^#^, Anomalies detected on prenatal USG; ^§^, Anomalies detected only on postnatal USG.

**Table 2 diagnostics-13-03106-t002:** Comparison of clinical course and abnormal findings according to degree of hydronephrosis.

	VUR *n* (%)		UPJO *n* (%)		Spontaneous Resolution *n* (%)		UTI *n* (%)		Surgery *n* (%)	
	Prenatal ^#^	Postnatal ^§^	Prenatal ^#^	Postnatal ^§^	Prenatal ^#^	Postnatal ^§^	Prenatal ^#^	Postnatal ^§^	Prenatal ^#^	Postnatal ^§^
Mild (7–9.9 mm)	2	1	0	1	22	18	4	3	0	0
Moderate (10–15 mm)	3	1	3	0	10	4	6	1	3	1
Severe (>15 mm)	5	2	5	0	2	1	9	2	7	2
Total	10	4	8	1	34	23	19	6	10	3

VUR, vesicoureteral reflux; UPJO, ureteropelvic junction obstruction; UTI, urinary tract infection; ^#^, Cases detected on prenatal USG; ^§^, Cases detected only on postnatal USG.

## Data Availability

The data that support the findings of this study are available from the corresponding author upon reasonable request.

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
