# Peer review of "The Diagnostic Efficacy of and Requirement for Postnatal Ultrasonography Screening for Congenital Anomalies of the Kidney and Urinary Tract"

_diagnostics, 2023, doi:10.3390/diagnostics13193106_

Round 1

Reviewer 1 Report

This study reported the outcomes of postnatal ultrasound screening of the kidney and urinary tract for 2620 infants. They found 4.62% of congenital anomalies of the kidney and urinary tract (CAKUT). A certain percentage of CAKUT can be detected by postnatal screening but not in prenatal screening. They concluded that postnatal ultrasonography screening of all children 27 for urinary tract anomalies would be beneficial.

Contrary to prenatal US screening, postnatal screening of CAKUT for all infants has not been incorporated into clinical practice. Because CAKUT is the main cause of kidney insufficiency in children, finding CAKUT is clinically significant.

For this study, it seems the novel point is unclear. Many papers have already reported the outcomes of postnatal screening on a larger scale.

  • 10.1016/0140-6736(91)92385-f

  • 10.1007/s00467-011-2098-0

and more literature cited by authors. The authors should also cite and discuss several papers that negatively support their ideas.

One crucial point to be revised is that the authors mixed the infant with or without a history of prenatal urological abnormality. Because of the study design, infants with prenatal CAKUT findings are more likely to receive postnatal US. As the authors described in the introduction, they should exclude or separately analyze the children who don't revive prenatal US or who have CAKUT at prenatal US.

To test the difference between unilateral vs. bilateral hydronephrosis is irrelevant to the aim of the study, and should be avoided.

  •  
  •  

The text is generally OK but some grammatical errors exist. It needs proofreading.

Author Response

Dear Editor,

I would like to thank editor and reviewers for their valuable comments on the article entitled “The Diagnostic Efficacy of and Requirement for Postnatal Ultrasonography Screening for Congenital Anomalies of the Kidney and Urinary Tract”.  Necessary corrections were made according to the comments of the reviewer and were made a list of point-by-point responses to the comments. Sincerely

Reviewers’ comments:

Reviewer 1:

This study reported the outcomes of postnatal ultrasound screening of the kidney and urinary tract for 2620 infants. They found 4.62% of congenital anomalies of the kidney and urinary tract (CAKUT). A certain percentage of CAKUT can be detected by postnatal screening but not in prenatal screening. They concluded that postnatal ultrasonography screening of all children 27 for urinary tract anomalies would be beneficial.

Contrary to prenatal US screening, postnatal screening of CAKUT for all infants has not been incorporated into clinical practice. Because CAKUT is the main cause of kidney insufficiency in children, finding CAKUT is clinically significant.

For this study, it seems the novel point is unclear. Many papers have already reported the outcomes of postnatal screening on a larger scale.

  • 1016/0140-6736(91)92385-f
  • 1007/s00467-011-2098-0

and more literature cited by authors. The authors should also cite and discuss several papers that negatively support their ideas.

Response: Several articles that did not support our ideas were cited and discussed.

One crucial point to be revised is that the authors mixed the infant with or without a history of prenatal urological abnormality. Because of the study design, infants with prenatal CAKUT findings are more likely to receive postnatal US. As the authors described in the introduction, they should exclude or separately analyze the children who don't revive prenatal US or who have CAKUT at prenatal US.

Response: The cases were divided into two groups as those with CAKUT detected on prenatal USG scan and those with CAKUT detected only on postnatal USG scan.

To test the difference between unilateral vs. bilateral hydronephrosis is irrelevant to the aim of the study, and should be avoided.

Response: Comparison of unilateral and bilateral hydronephrosis was excluded from the text.

  • Comments on the Quality of English Language

The text is generally OK but some grammatical errors exist. It needs proofreading.

Response: The English version of the article has been revised.

Reviewer 2 Report

Thank you for your very nice paper.

Just a few comments regarding the structure of the paper. The conclusion part should be a separate chapter, like the results or introduction.

The limits of the study should be discussed in more detail, since they measure the real value of your work.

The ultrasonography machine should be described as per the current guidelines, including, in brackets, the city and country of the manufacturer.

I also suggest you do another English proofreading, since there are a few typos worth addressing.

Author Response

Dear Editor,

I would like to thank editor and reviewers for their valuable comments on the article entitled “The Diagnostic Efficacy of and Requirement for Postnatal Ultrasonography Screening for Congenital Anomalies of the Kidney and Urinary Tract”.  Necessary corrections were made according to the comments of the reviewer and were made a list of point-by-point responses to the comments. Sincerely

Reviewer 2:

Thank you for your very nice paper.

Just a few comments regarding the structure of the paper. The conclusion part should be a separate chapter, like the results or introduction.

Response: The conclusion section has been corrected as a separate chapter.

The limits of the study should be discussed in more detail, since they measure the real value of your work.

Response: The limits of the study were discussed in more detail.

The ultrasonography machine should be described as per the current guidelines, including, in brackets, the city and country of the manufacturer.

Response: The city and country of the manufacturer were added in the bracket.

I also suggest you do another English proofreading, since there are a few typos worth addressing.

Response: The English version of the article has been revised.

Round 2

Reviewer 1 Report

The authors provided sufficient revision.